# A land surface model combined with a crop growth model for paddy rice (MATCRO-Rice Ver. 1) – Part II: Model validation

Yuji Masutomi[1], Keisuke Ono[2], Takahiro Takimoto[3], Masayoshi Mano[4], Atsushi Maruyama[2], and Akira Miyata[2]

[1]College of Agriculture, Ibaraki University, 3-21-1, Chuo, Ami, Inashiki, Ibaraki 300-0393, Japan
[2]Intstitute for Agro-Environmental Sciences, NARO, 3-1-3, Kannondai, Tsukuba, Ibaraki 305-8604, Japan
[3]Institute for Global Change Adaptation Science, Ibaraki University, 3-21-1, Chuo, Ami, Inashiki, Ibaraki 300-0393, Japan
[4]Graduate School of Horticulture, Chiba University, 648 Matsudo, Matsudo-shi, Chiba 271-8510, Japan

*Correspondence to:* Yuji Masutomi (yuji.masutomi@gmail.com)

**Abstract.** We conducted two types of validation for the simulations by MATCRO-Rice developed by Masutomi et al. (2016). In the first validation, we compared simulations with observations for latent heat flux (LHF), sensible heat flux (SHF), net carbon uptake by crop, and paddy rice yield from 2003 to 2006 at the site where model parameters are parameterised. In the second validation, we compared the observed and simulated paddy rice yields over Japan from 1991 to 2010 between observations and simulations. The 4-year average root mean square errors (RMSEs) of the first validation for LHF and SHF were 18.20 and 15.47 W m$^{-2}$, respectively. These values for errors are comparable to those reported in earlier studies. The comparison of biomass growth during growing periods from 2003 to 2006 at the parameterisation site shows that the simulations were in agreement with the observations, indicating that the model can reproduce the net carbon uptake by crop well. The 4-year average RMSE of the first validation for crop yield in the same period was 410.6 kg ha$^{-1}$, which accounted for 8.1% of the mean observed yields. The error of the second validation for crop yield was 16.7% and the correlation of crop yields between observations and simulations from 1991 to 2010 was significant at 0.663 (P<0.01). These results indicate that MATCRO-Rice has high ability to accurately and consistently simulate LHF, SHF, net carbon uptake by crop, and crop yield.

## 1 Introduction

It has been recognized that crop growth and management in agricultural land are important factors that affect climate at various spatial and temporal scales via exchange of heat, water, and gases (Tsvetsinskaya et al., 2001;Bondeau et al., 2007; Osborne et al., 2009; Levis et al., 2012). Betts (2005) pointed out that integration of crop growth models (CGMs) into climate models is needed for accurate climate simulations by climate models. To consider the influence of agricultural land on climate in climate simulations, several land surface models (LSMs) or dynamics vegetation models (DVMs) incorporated with a CGM have been developed (Tsvetsinskaya et al., 2001; Kucharik, 2003; Gervois et al., 2004; Bondeau et al., 2007; Osborne et al., 2007; Lokupitiya et al., 2009; Maruyama and Kuwagata, 2010; Levis et al., 2012; Osborne et al., 2015).

Masutomi et al. (2016) have developed a new LSM-CGM combined model, called MATCRO-Rice, by incorporating a CGM into a LSM, MATSIRO (Takata et al., 2003). The most important feature of the model is that it can consistently simulate latent heat flux (LHF), sensible heat flux (SHF), net carbon uptake by crop, and crop yield in paddy rice fields by exchanging variables between an LSM and a CGM. The consistency among model outputs enable us to apply the model to a wide range of integrated issues. For example, the model can investigate the interaction between climate and paddy rice fields, consistently considering impacts of climate on rice productivity and impacts of paddy rice fields on climate. Osborne et al. (2009) showed that this interaction can affect variability in climate and crop production. Therefore, the understanding of the interaction is important for securing food security. However, little is known about the interaction. MATCRO-Rice can be a useful tool to study the interaction between climate and paddy rice fields.

The objective of the present paper is to present the results of the comprehensive validation of MATCRO-Rice and to show the effects of modifications from the original LSM, MATSIRO. Before presenting the results of the validation and the effects of modification, we first show the numerical method (Section 2) and the method and results of parameterisation for model parameters (Section 3). The results of model validation and the effects of modifications are shown in Sections 4 and 5, respectively, followed by concluding remarks in Section 6

## 2 Numerical setting and method

All simulation setting parameters are shown in Table 1. We set the time resolution of the simulation to half hour, i.e., $\delta_t = 1800$ seconds. For time discretisation, the forward difference method was used.

To simulate soil water and heat transfer (Section 3.5 in Masutomi et al. (2016)), we spatially discretised soil into five layers with thickness of 0.05, 0.2, 0.75, 1.0, and 2.0 m, resulting in $z_{max} = 4.0$ m, $z_t = 0.05$ m, and $z_b = 2.0$ m. To simulate soil water content for each soil layer ($w_s$), we replaced the gradient of water flux by net water fluxes between layers. In the calculation for water fluxes between layers, we used the hydraulic conductivity that is smaller among soil layers and the difference in water potentials between soil layers. After the calculation for soil water content for each layer, water content beyond saturation was taken out to base flow.

To simulate soil temperature for each soil layer, we solved the system of equations for soil layers by using the Gauss-Jordan method. In the calculation of soil temperatures, we replaced the gradient of heat flux by net heat fluxes between layers. In the calculation of heat fluxes between layers, we used thermal conductivity averaged between soil layers and soil temperatures for each layer.

The downhill simplex method (Nelder and Mead, 1965) was used to simulate temperatures of the canopy and surface ($T_c$ and $T_g$; Section 3.1 in Masutomi et al. (2016)), bulk transfer coefficients ($C_{Eg}$, $C_{Ec}$, $C_{Hg}$, $C_{Hc}$, $C_M$, and $C_{Mg}$; Section 3.3 in Masutomi et al. (2016)), and variables related to carbon assimilation ($\overline{A}_{n,x}$, $c_{i,x}$, and $\overline{g}_{st,x}$; Section 4.1 in Masutomi et al. (2016)) .

We set $z_a = 3$ m. $CO_2$ concentration ($C_{a,ppm}$) and the depth of surface water ($d_w$) were set at 390 ppm and 0.025 m, respectively. The initial dry weight of each organ was set at 1 kg ha$^{-1}$ for leaf ($W_{lef,0}$), stem ($W_{stm,0}$), and root ($W_{rot,0}$) and at 0.5 kg ha$^{-1}$ for glucose reserve in leaf ($W_{glu,0}$).

$D_{oy,Ie}$, $D_{oy,Is}$, $D_{oy,sw}$, and $L_t$ depend on the simulations. Values for these parameters are shown in the sections of each simulation.

## 3  Parameterisation

Table 2 shows model parameters parameterised in the present paper. All parameters are parameterised using observations, the literature, and assumptions. The method of the parameterisation is explained in this section.

### 3.1  Parameterisation site and observation data

Table 3 shows the observational data used for parameterisation. The data were observed from 2003 to 2006 at a site which is located in Tsukuba, Japan (Lat: 36° 03' 14.3" N; Lon: 140° 01' 36.9" E), at 13 m above sea level. The climatic zone of the site is temperate, with the mean annual air temperature 13.7°C and precipitation 1200 mm. The soil type is clay loam. The variety planted at the site is "Koshihikari", which is the most planted variety in Japan.

Biomass for each organ ($W_{lef}$, $W_{pnc}$, $W_{rot}$, and $W_{stm}$) and leaf area index ($L$) were measured nearly every two weeks. At each measuring time, ten stands were sampled from the fields. Yield ($Y_{ld}$) and phenological dates including transplanting ($D_{oy,tr}$), heading ($D_{oy,hd}$), and harvest ($D_{oy,hv}$) were observed every year. The values of observed yield are the husked rice yield with 15% water content. The rice grains for measuring yield were sampled from the whole fields of the observational site. The crop height ($h_{gt}$) was measured on average every 5 days.

### 3.2  Phenology

Phenological parameters that represent development stages ($D_{vs,e}$, $D_{vs,h}$, $G_{ds,m}$, $D_{vs,tr}$, and $D_{vs,te}$) were parameterised. First, we calculated $D_{vs}$s at heading and $G_{ds,m}$s from 2003 to 2006 using the phenological model given by Masutomi et al. (2016). The mean values were set to $D_{vs,h}$ and $G_{ds,m}$, resulting in $D_{vs,h} = 0.616$ and $G_{ds,m} = 167759940$ K·s. Figure 1 compares the heading and harvest dates between observations and simulations from 2003 and those from 2006. The simulated heading and harvest dates were in good agreement with the observations. The average errors were 2.25 and 4.5 days for heading and harvest, respectively.

$D_{vs,e}$, $D_{vs,tr}$, and $D_{vs,te}$ were determined so that the duration from sowing to emergence, transplanting, and the end of transplanting shock was 5, 20, and 25 days, respectively. Thus, $D_{vs,e} = 0.012$, $D_{vs,tr} = 0.06$, and $D_{vs,te} = 0.08$.

### 3.3  Partitioning

MATCRO partitions carbohydrates in leaves, in the form of glucose, into each organ, according to MACROS (Penning de Vries et al., 1989). Parameters related to glucose partitioning ($D_{vs,rot1}$, $D_{vs,rot2}$, $D_{vs,lef1}$, $D_{vs,lef2}$, $D_{vs,pnc1}$, $D_{vs,pnc2}$, $P_{rot}$, and $P_{lef}$)

were parameterised as follows: (i) we calculated the ratio of glucose partitioned to each organ (leaf, stem, root, panicle) during the growing period using the observed biomass for each organ; (ii) we conducted the curve fitting of the calculated ratios in (i). Figure 2 shows the calculated ratios of glucose partitioned to each organ and the fitting curves for the ratios.

To determine the ratio of dead leaf at harvest ($r_{\mathrm{d1,lef}}$), we first calculated the observational ratios of dead leaf during growing period by dividing the decrease in leaf biomass between observational dates by the duration among the observational dates. Then by graphically fitting a curve to the calculated ratios of dead leaf, we determined $r_{\mathrm{d1,lef}}$. Figure 3 shows the calculated ratios of dead leaf and the fitted curve.

The fraction of glucose allocated to starch reserve ($f_{\mathrm{stc}}$) is determined as follows: (i) we first calculated the ratios of stem biomass at harvest to maximum stem biomass for each year from 2003 to 2006 (Bouman et al. (2001)); (ii) then, a 4-year average was calculated for $f_{\mathrm{stc}}$.

## 3.4 LAI, crop height, and specific leaf weight

To obtain the parameters for the relationship between LAI and crop height ($h_{\mathrm{aa}}, h_{\mathrm{ab}}, h_{\mathrm{ba}},$ and $h_{\mathrm{bb}}$), we conducted linear regressions of the data before and after heading using observations for LAI and crop height from 2003 to 2006. Thus, $h_{\mathrm{aa}} = 0.439, h_{\mathrm{ab}} = 0.675, h_{\mathrm{ba}} = 0.366,$ and $h_{\mathrm{bb}} = 0.318$. Figure 4 compares the LAI–height relation between observations and simulations.

To obtain parameters for specific leaf weight ($k_{\mathrm{S_{lw}}}, S_{\mathrm{lw,mn}},$ and $S_{\mathrm{lw,mx}}$), we plotted observations for specific leaf weights during growing periods from 2003 to 2006 and conducted the curve fitting of the plotted data. Thus, $k_{\mathrm{S_{lw}}} = 3.5, S_{\mathrm{lw,mn}} = 350,$ and $S_{\mathrm{lw,mx}} = 600$. Figure 5 shows the specific leaf weights and the fitted curve.

## 3.5 Crop yield

To determine the ratio of crop yield to dry weight of panicle at harvest ($k_{\mathrm{yld}}$), we calculated the dry weight of panicle at harvest, because the weight was not observed. By assuming linear increase of dry weight from the last date in which dry weight of the panicle was measured, we calculated the dry weight of the panicle at harvest from 2003 to 2006. The median value among the ratios of observed yields to the calculated dry weight of panicle produced $k_{\mathrm{yld}}$.

## 3.6 Rubisco-limited photosynthesis rate

Parameters related to Rubisco-limited photosynthesis rate ($V_{\mathrm{max}}(0), s_1,$ and $s_2$) were parameterised using the values obtained from the literature. In this parameterisation, we adjusted the parameters so that the Rubisco-limited photosynthesis rate ($\overline{\omega}_{\mathrm{c},x}$) simulated by MATCRO agrees with the observational value reported by Borjigidai et al. (2006). In the simulations, $CO_2$ concentration in the leaf was fixed to $c_{\mathrm{i},x} = 30$ Pa. Figure 6, showing the comparison of Rubisco-limited photosynthesis rates among MATCRO, those reported by Borjigidai et al. (2006), and MATSIRO (Takata et al., 2003), on which MATCRO is based, indicate that there is a good agreement in the photosynthesis rate between the simulations of MATCRO and the observational

value in Borjigidai et al. (2006); the simulations for the photosynthesis rate of MATCRO were significantly improved compared to those of MATSIRO.

## 4  Validation

We conducted two types of validation. The first validation was conducted at the parameterisation site as explained in Section 3.1. In the validation, the simulated LHF, SHF, carbon uptake by crop, and crop yields were compared with the observations from 2003 to 2006. The second validation was conducted for a territory across Japan. The simulated crop yields for Japan were compared with the national statistics from 1991 to 2010.

### 4.1  Validation at the parameterisation site

#### 4.1.1  Input and validation data

Table 4 shows the observational data used for the validation. Information on the instruments used for the observations are available from the AsiaFlux web site (AsiaFlux, 2016). The height at which the wind speed was measured was different each year. Assuming logarithmic vertical profile of the wind, we transformed the observed wind speed to that at 3 m above ground, because the reference height ($z_a$) is set to be 3m (Section 3.1). It is noted that we used the observed values of photosynthetic active radiation (PAR) in addition to the standard meteorological inputs, because PAR, which is often not measured, was observed at the parameterisation site. We set $L_t = 36.05$. Values for soil parameters for clay loam are shown in Table 5. The "Koshihikari" variety was planted using a transplanting technique. We set $D_{oy,Is}$=114, 107, 114, 113, and $D_{oy,Ie}$=231, 251, 243, 241 from 2003 to 2006, respectivey, using the observations for the depth of surface water ($d_{wo}$). $D_{oy,sw}$ was calculated from the observed transplanting date ($D_{oy,tr}$), assuming that transplanting was conducted 20 days after sowing, i.e., $D_{oy,sw} = D_{oy,tr} - 20$.

The validation was conducted from 2003 to 2006, although the AsiaFlux provides the observational data from 2001 to 2006 for the site. We did not use the observational data before 2002 because the flux tower was relocated in the paddy fields in April 2003. Thereafter, the observed flux data have been more representative of the field, where the rice sampling was conducted.

#### 4.1.2  Comparison of LHF and SHF

Figures 7 to 10 show the comparison of the daily and half-hourly LHF and SHF from 2003 to 2006. We can observe that MATCRO can replicate the daily and half-hourly variations in LHF and SHF accurately. Quantitatively, the RMSEs of daily LHF between simulations and observations for each year were 15.15, 21.84, 17.25, and 18.57 W m$^{-2}$, with the 4-year average of 18.20 W m$^{-2}$ (Table 6). The RMSEs of daily SHF were 13.62, 14.72, 14.84, and 18.69 W m$^{-2}$, with the 4-year average of 15.47 W m$^{-2}$ (Table 7). These RMSE values are comparable to those reported in earlier studies (Kimura and Kondo,1998; Maruyama and Kuwagata,2010).

One of the major reasons for the errors of LHF and SHF between simulations and observations is thought to be a problem in flux observations. Aubinet et al. (2000) reported that the energy balance in observations is not closed. In contrast, the energy balance simulated by MATCRO is completely closed. Therefore, the energy imbalance in flux observations can cause errors between simulations and observations. El Maayar et al. (2008) suggested to test the degree of energy imbalance in observations before comparing the observations with simulations. This degree is generally evaluated by

$$I_{\mathrm{m}} = \left( \sum_d \frac{(\overline{H}(d) + \lambda \overline{E}(d))}{(\overline{R}_{\mathrm{n}}(d) - \overline{G}(d))} \right) / N, \tag{1}$$

where $\overline{H}$, $\lambda \overline{E}$, $\overline{R}_{\mathrm{n}}$, and $\overline{G}$ are the daily averages for SHF, LHF, net radiation, and heat flux into ground, respectively, $d$ indicates a day, and $N$ is the number of days. The observation values for $R_{\mathrm{n}}$ and $G$ in this equation are expected to be sufficiently accurate (Twine et al., 2000; Wilson et al., 2002). The values of $I_{\mathrm{m}}$ in the observations from 2003 to 2006 were 0.79, 0.77, 0.78, and 0.74, with the average of 0.78. In other words, these results imply that the total flux of observed LHF and SHF can be smaller than a true value. The ratio of the total flux of observed LHF and SHF to that of simulated LHF and SHF from 2003 to 2006 were 0.84, 0.79, 0.80, and 0.83, with the average of 0.82. This suggests that the errors of LHF and SHF between observations and simulations can be largely attributed to the energy imbalance in observations.

### 4.1.3 Comparison of net carbon uptake by crop

In this section, we tested the accuracy of MATCRO for simulating net carbon uptake by crop during growing periods by comparing the changes in total biomass between simulations and observations. Figure 11 compares the growth of the total biomass between simulations and observations from 2003 to 2006. As indicated by the figure, the simulated total biomass was in good agreement with the observations. Hence, we conclude that the model has high accuracy for simulating net carbon uptake by crop during growing period.

### 4.1.4 Comparison of yield

Figure 12 shows the comparison of the observed and simulated yields from 2003 to 2006. As indicated by the figure, MATCRO can reproduce well the absolute values of crop yields. The mean RMSE from 2003 to 2006 was 410.6 kg ha$^{-1}$, which was 8.1% of the mean observed yields. However, the model overestimated the crop yields in 2003. The primary cause of the large overestimation in 2003 can be attributed to the late harvest in the simulation for 2003; the model delayed the harvest by 11 days in 2003 (see Section 3.2). To confirm this, we recalculated the yield in 2003 by using the observed harvest date. The revised yield is shown in the figure as a red circle. The revised yield was in good agreement with the observations in 2003. These results suggest that the phenological model in MATCRO should be further improved for a more accurate estimation of crop yield. The current version of the phenological model in MATCRO implements only the temperature. The consideration of the photoperiod may further improve the accuracy of the phenological model in the simulation of harvest date as well as heading date (e.g., Penning de Vries et al., 1989; Connor et al. ,2011).

### 4.2 Validation over Japan

#### 4.2.1 Method, input and validation data

Rice yields for all prefectures in Japan were simulated from 1991 to 2010, and then the average national rice yield was calculated for each year from the prefectural yields weighted by the prefectural planting areas. The simulated rice yields for Japan were compared with the national crop statistics (MAFF, 1991-2010).

Global Meteorological Forcing Dataset (GMFD) for land surface modelling (Sheffield et al., 2006) was used for meteorological input data. Because the spatial resolution of the GMFD is 1 degree, the average meteorological values for each prefecture were calculated from the gridded values weighted by the prefectural planting areas. The time resolution of the GMFD is 3 hours. We used the same values at each 3 hours for half-hourly simulations. The comparison with the agro-meteorological dataset in Japan (Ohno et al., 2002) revealed that shortwave radiation of the GMFD has a large error; we corrected the error using the agro-meteorological dataset, so that the daily values for shortwave radiation of the GMFD could agree with those of the agro-meteorological dataset. In addition, we changed the ratio of photosynthetic active radiation to shortwave radiation from 0.5 (default) to 0.423, according to the value observed in the parameterisation site (Section 3.1).

The sowing and harvesting dates ($D_{\mathrm{oy,sw}}$ and $D_{\mathrm{oy,hv}}$) for each prefecture were obtained from the national crop statistics (MAFF, 1991-2010). In this validation, simulated harvesting dates were not used because the results in the previous section (Section 4.1.4) showed that simulated harvesting dates may cause errors in yield simulations.

For simplicity, land surface was assumed to be annually flooded and irrigated. Hence we set $D_{\mathrm{oy,Is}} = 1$ and $D_{\mathrm{oy,Ie}} = 365$. $L_{\mathrm{t}}$ for each prefecture was set to the latitude of the center of each prefecture. The other simulation settings and parameters for crop and soil were set using the values shown in Tables 1, 2, and 5.

#### 4.2.2 Comparison of yield

Figure 13 shows the comparison of the observed and simulated yields from 1991 to 2010. The average error between simulations and observations was 16.7%. Although the simulated yields tended to overestimate the observations, the correlation between simulations and observations was significant at 0.663 (P<0.01). Hence, we conclude that MATCRO can reproduce substantial parts of weather-induced variability in yields. One of the reasons for the simulation errors is thought to be the method of parameterisation. For the simulations over Japan, we used parameters parameterised for a variety "Koshihikari" only at one site in Japan (Section 3), although there is a large diversity in agricultural management and technique, and rice varieties planted throughout Japan. Therefore, parameterisation at a large scale is necessary for better large scale simulations.

## 5 Effects of modifications

There are two major modifications of MATCRO from the original LSM (MATSIRO). The first one is the dynamic calculation of LAI, crop height, and root. The other is the consideration of flooded surface and irrigation. We quantify the effects of the

two major modifications on the simulation of LHF and SHF. Both simulations are conducted at the parameterisation site from 2003 to 2006 (Section 3.1).

## 5.1 Effect of dynamic calculation of LAI

The original LSM (MATSIRO) uses the monthly constant LAI, which is given in grids by grids. The default gridded LAI data
of MATSIRO were obtained from the Global Soil Wetness Project 2 (Dirmeyer et al., 2006). Figure 14 shows the comparison of LAI between observations, simulations by MATCRO, and the default values of MATSIRO. We can see that MATCRO reproduces adequately seasonal changes in LAI well, although the default LAI are not in agreement with the observed LAI. MATSIRO also uses constant crop height and root length, which are vegetation-specific parameters. The default values of crop height and root length for crops are 1m. Using the default data for LAI and the default values for crop height and root
length, we simulated LHF and SHF from 2003 to 2006. In the simulations, we also used the original equation of MATSIRO for calculating the maximum canopy water ($w_{cap}$).

The RMSEs of daily LHF from 2003 to 2006 were 19.4, 20.78, 19.28, and 18.72 W m$^{-2}$, respectively, with the 4-year average of 19.54 W m$^{-2}$ (Table 6). The RMSEs of daily SHF from 2003 to 2006 were 14.69, 20.30, 16.93, 20.68 W m$^{-2}$, respectively, with the 4-year average of 18.15 W m$^{-2}$ (Table 7). These errors are compatible to those of MATCRO. Hence, we
conclude that the effects of the dynamic calculation of LAI, crop height and root length on LHF and SHF are small.

## 5.2 Effect of flooded and irrigated surface

We simulated LHF and SHF from 2003 to 2006 without flooded surface and irrigation. The simulations are called MATCRO-RF, which denotes simulations under rain-fed conditions. Figures 15 and 16 show the comparison of daily LHF and LHF between observations and simulations obtained by MATCRO and MATCRO-RF. The LHF and SHF simulated by MATCRO-
20 RF were not in agreement with the observations. The RMSEs of daily LHF simulated by MATCRO-RF from 2003 to 2006 were 16.63, 36.90, 29.32, and 24.93 W m$^{-2}$, respectively, with the 4-year average of 26.95 W m$^{-2}$ (Table 6). The RMSEs of daily SHF from 2003 to 2006 were 16.34, 42.02, 34.16, 31.56 W m$^{-2}$, respectively, with the 4-year average of 31.02 W m$^{-2}$ (Table 7). These errors in MATCRO-RF are considerably larger than those simulated by MATCRO. Figures 15 and 16 show that MATCRO-RF tends to underestimate daily LHF and to overestimate daily SHF. The underestimation of daily LHF can be
attributed to lower evaporation from the soil due to the absence of irrigation and flooding. The overestimation of daily SHF can be caused by high surface temperature due to lower evaporation from the soil. Hence, we conclude that flooded surface and irrigation have large effects on simulations of LHF and SHF.

## 6 Concluding remarks

In this paper, we presented the results of the validation of MATCRO-Rice and the effects of the modification of the original LSM
(MATSIRO), and the numeric and parameterisation methods. First, the comparison of the LHF and SHF between simulations and observations at the paramerisation site confirmed that the model can reproduce the observed LHF and SHF data well. The

accuracy of the simulations for LHF and SHF was comparable to those obtained in earlier studies. Second, we showed that the simulated growth of the total biomass was in good agreement with the observations at the parameterisation site. This indicates that the model can simulate the net carbon uptake by crop during a growing period at paddy rice fields. Last, we demonstrated that the model has high ability to simulate crop yield by comparing the simulated and observed yields at the parameterisation site and over Japan.

The validation results suggest that MATCRO-Rice has high ability to accurately and consistently simulate LHF, SHF, net carbon uptake by crop, and yield. There have been many models that simulate some of the four variables with high accuracy, but a few models can accurately and consistently simulate all four of them. This point is the most important feature of MATCRO-Rice. The model can be applied to a wide range of issues, including climate change impact (e.g., Masutomi et al.,2009), and it will facilitate the scientific research especially on the climate–crop interactions (Osborne et al.,2009).

We validated LHF, SHF, and carbon flux simulated by this model with observations from only one site. The model should be further validated at multiple sites in order to enforce the reliability and applicability of the model. However, since there are a few flux sites on agriculture land worldwide, it will be necessary to increase their number on agricultural land to promote climate–crop modelling studies.

We assessed the effects of the dynamics simulation of LAI, crop height and root length on LHF and SHF and the effect of flooded surface and irrigation on LHF and SHF. The results show that the effects of the dynamic simulation on LHF and SHF are small, whereas the flooded surface and irrigation have large effects on LHF and SHF. These results suggest that climate–crop modelling should incorporate flooded surface and irrigation.

## 7 Code availability

The source code of MATCRO will be distributed at request to the corresponding author (Yuji Masutomi: yuji.masutomi@gmail.com). The website for MATCRO-Rice will be developed in the near future.

*Acknowledgements.* We are grateful to Mrs Hatanaka for her help in extensive literature survey. This research was supported by the Environment Research and Technology Development Fund (S-12) and the Program on Development of Regional Climate Change Adaptation Plans in Indonesia (PDRCAPI) of the Ministry of the Environment.

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

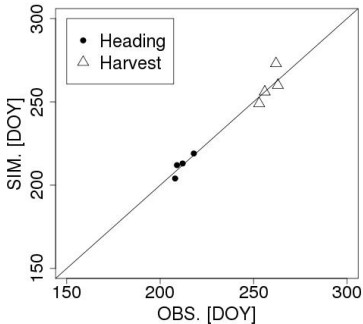

**Figure 1.** Comparison of heading and harvest dates. SIM: simulations; OBS: observations; DOY: The number of days from Jan. 1.

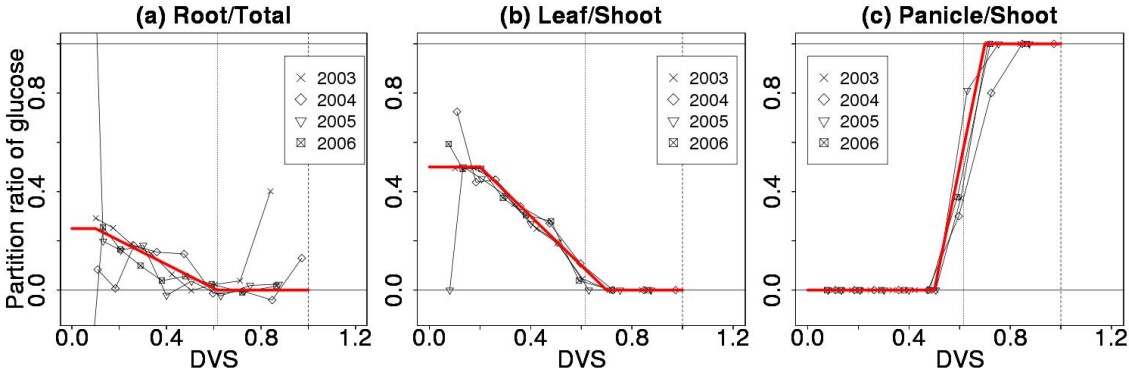

**Figure 2.** Partitioning ratio of glucose. Red lines are fitted. DVS: development stage

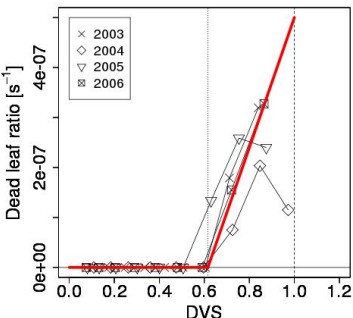

**Figure 3.** Ratio of dead leaf. A red line is fitted. DVS: development stage

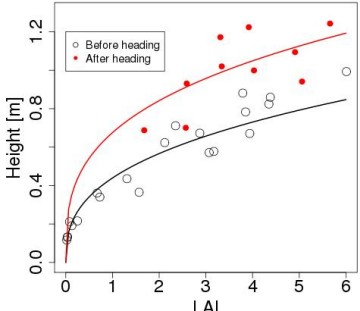

**Figure 4.** Relationship between the leaf area index (LAI) and crop height (black curve: $h_{\mathrm{gt}} = h_{\mathrm{aa}} L^{h_{\mathrm{ab}}}$; red curve: $h_{\mathrm{gt}} = h_{\mathrm{ba}} L^{h_{\mathrm{bb}}}$)

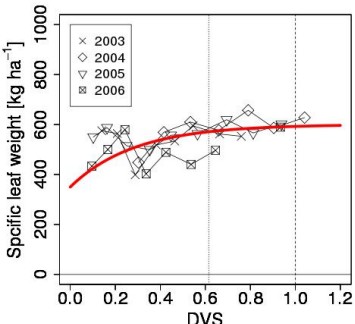

**Figure 5.** Relationship between specific leaf weight and development stage (DVS). A red curve denotes the fitted curve used in the model.

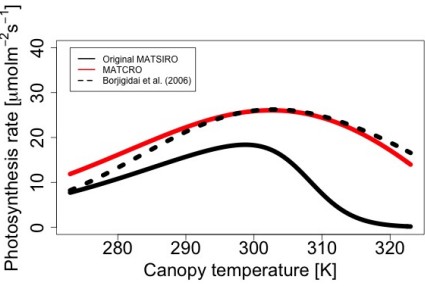

**Figure 6.** Rubisco-limited photosynthesis rate

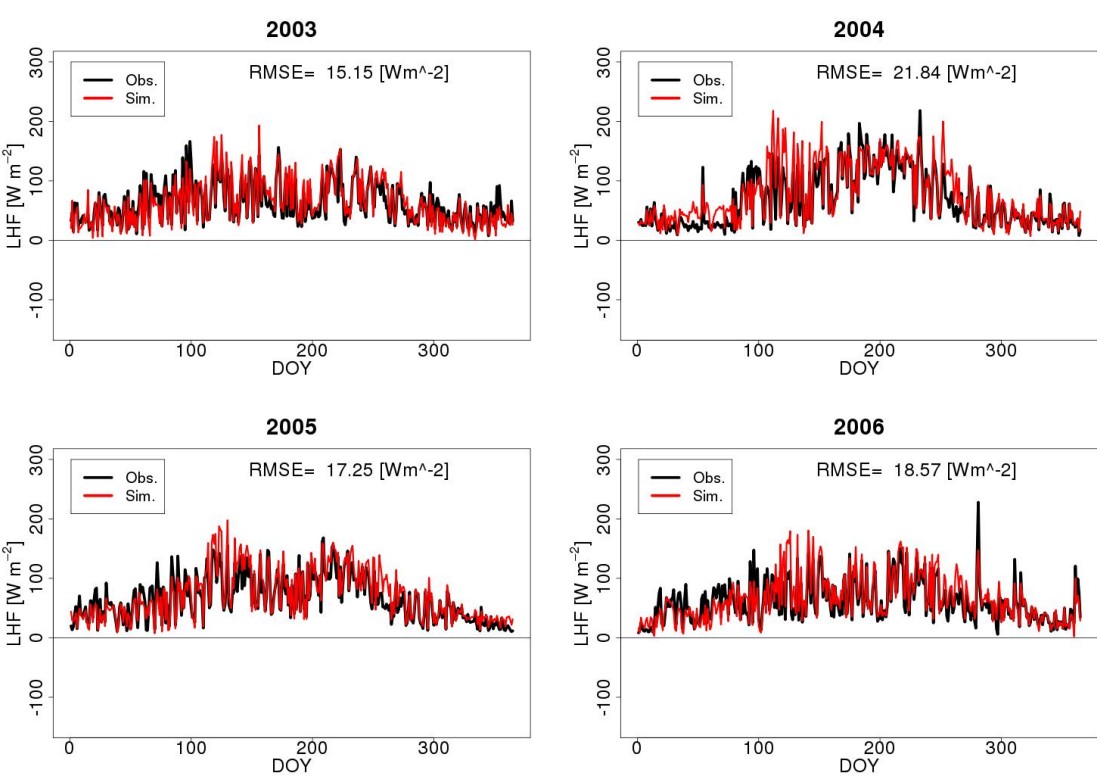

**Figure 7.** Comparison of daily latent heat flux (LHF) between simulations and observations. DOY: The number of days from Jan. 1.

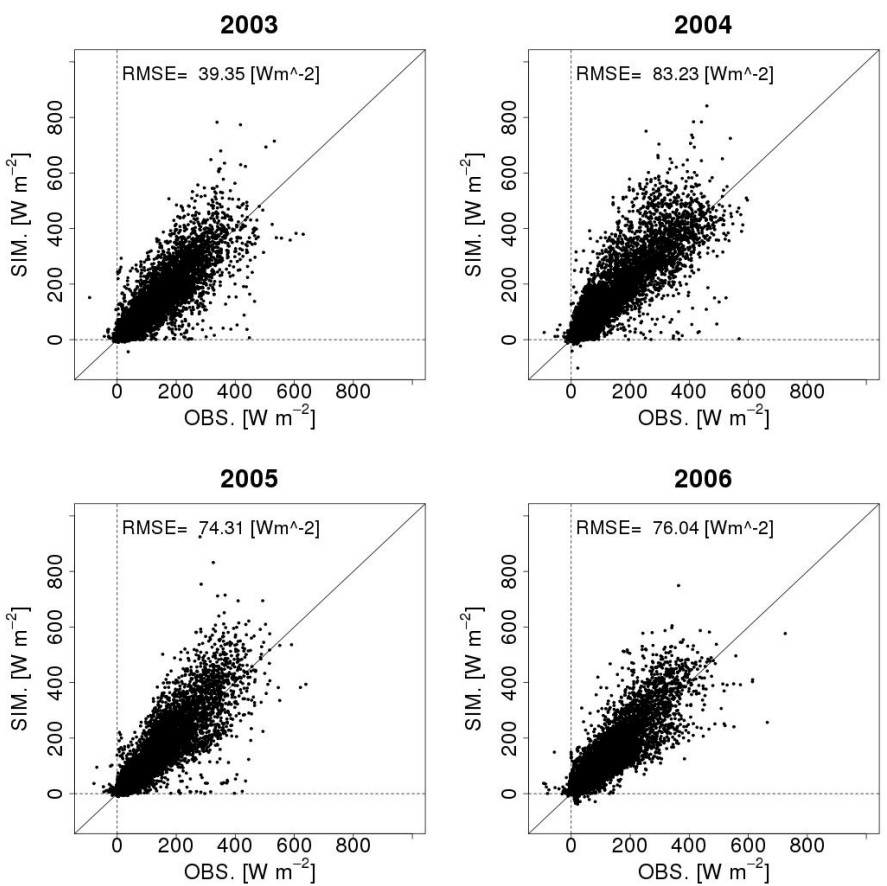

**Figure 8.** Comparison of half-hourly latent heat flux (LHF) between simulations and observations.

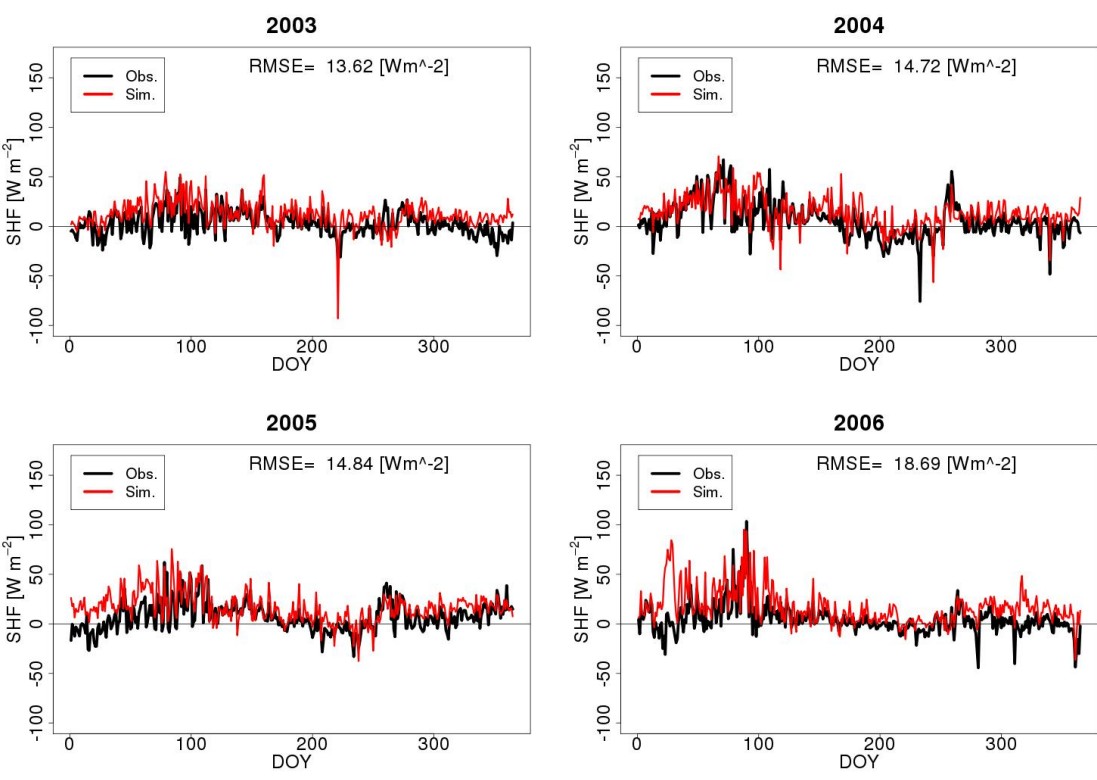

**Figure 9.** Comparison of sensible heat flux (SHF) between simulations and observations. DOY: The number of days from Jan. 1.

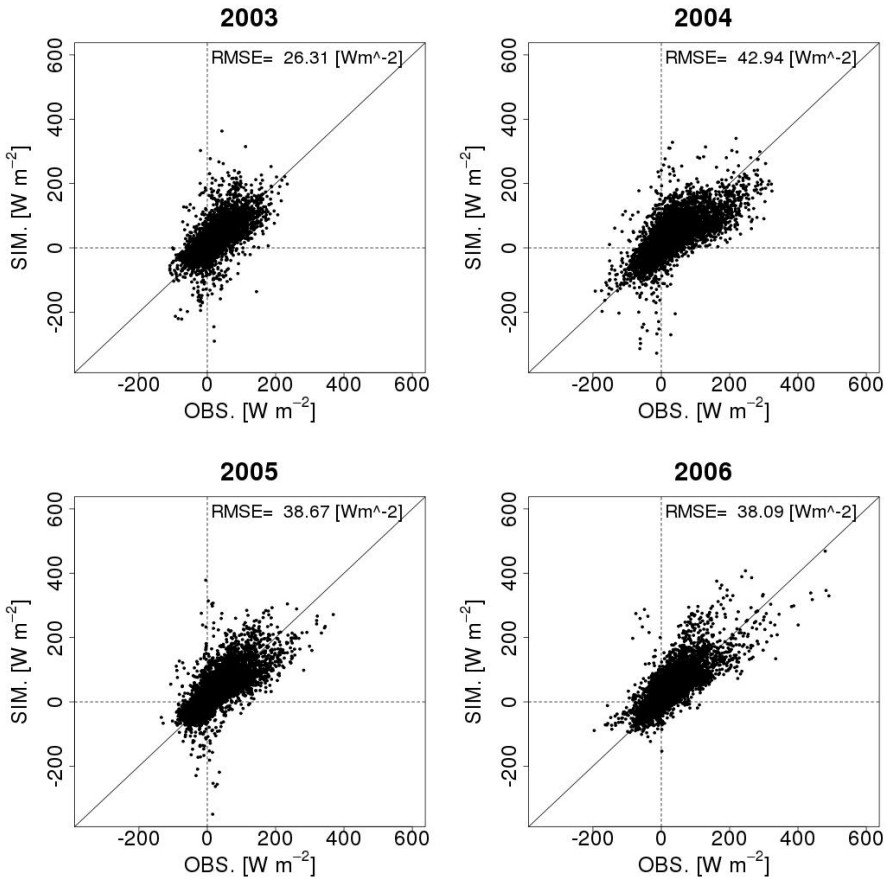

**Figure 10.** Comparison of half-hourly sensible heat flux (SHF) between simulations and observations.

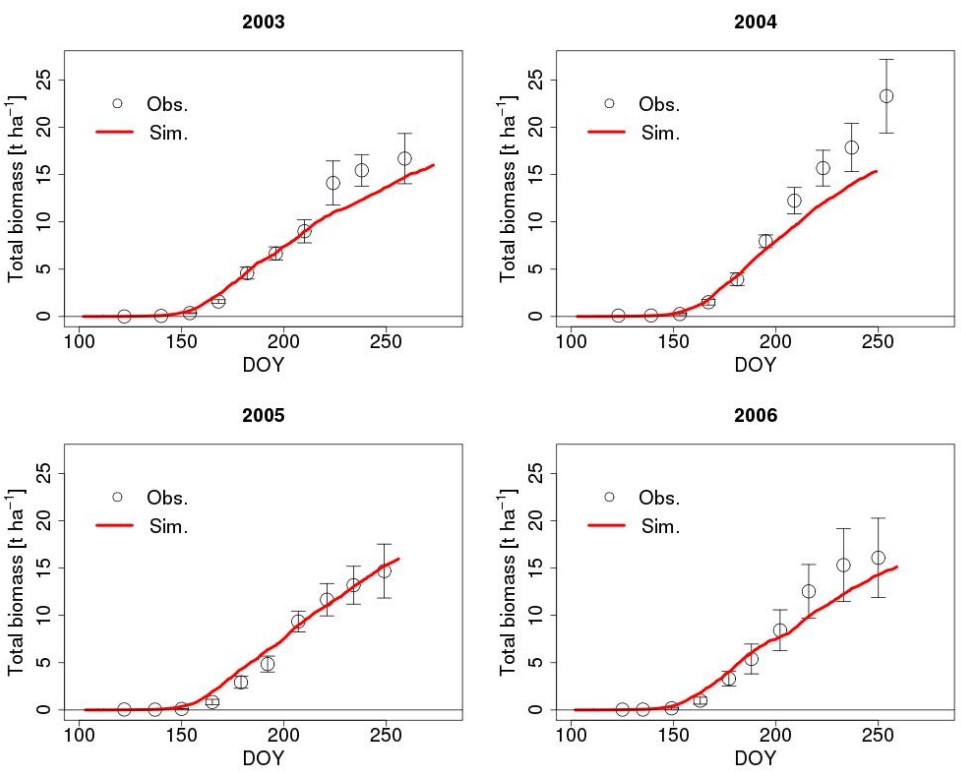

**Figure 11.** Comparison of total biomass between simulations and observations during growing periods from 2003 to 2006. Circles indicate mean values of observations and the ranges indicate standard deviation of observations. Red lines denotes simulations. DOY: The number of days from Jan. 1.

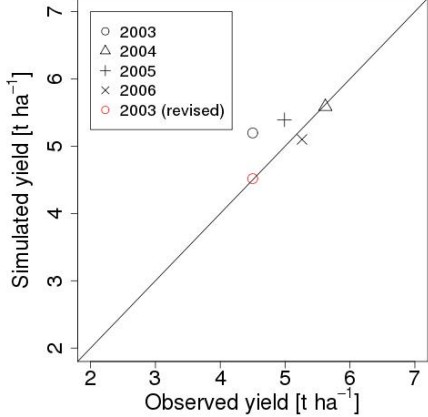

**Figure 12.** Comparison of yields between simulations and observations

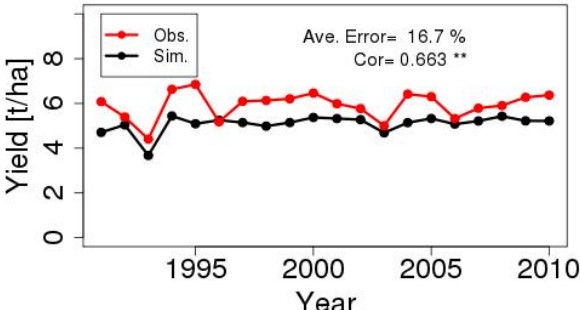

**Figure 13.** Comparison of yields over Japan between simulations and observations

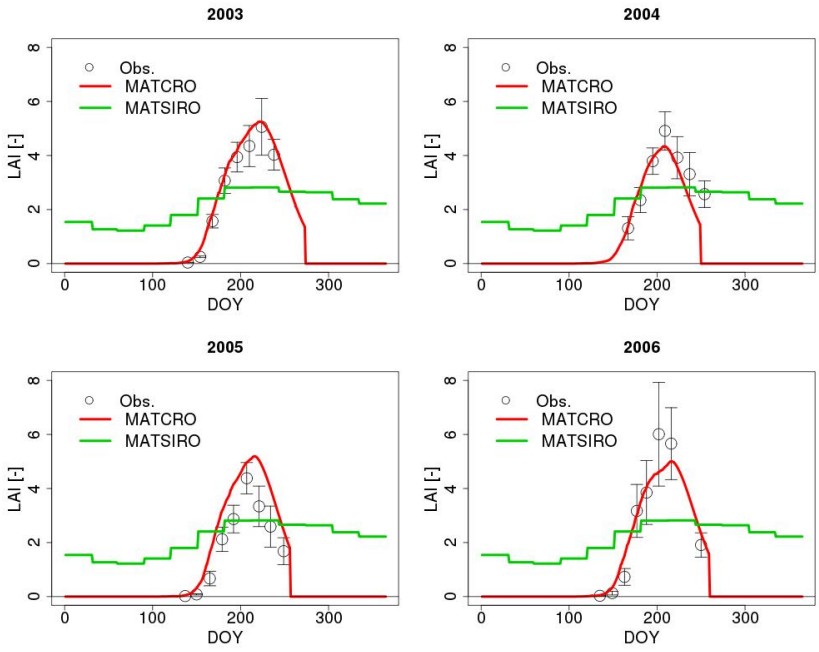

**Figure 14.** Comparison of LAI between observations, simulations by MATCRO, and the default value of MATSIRO

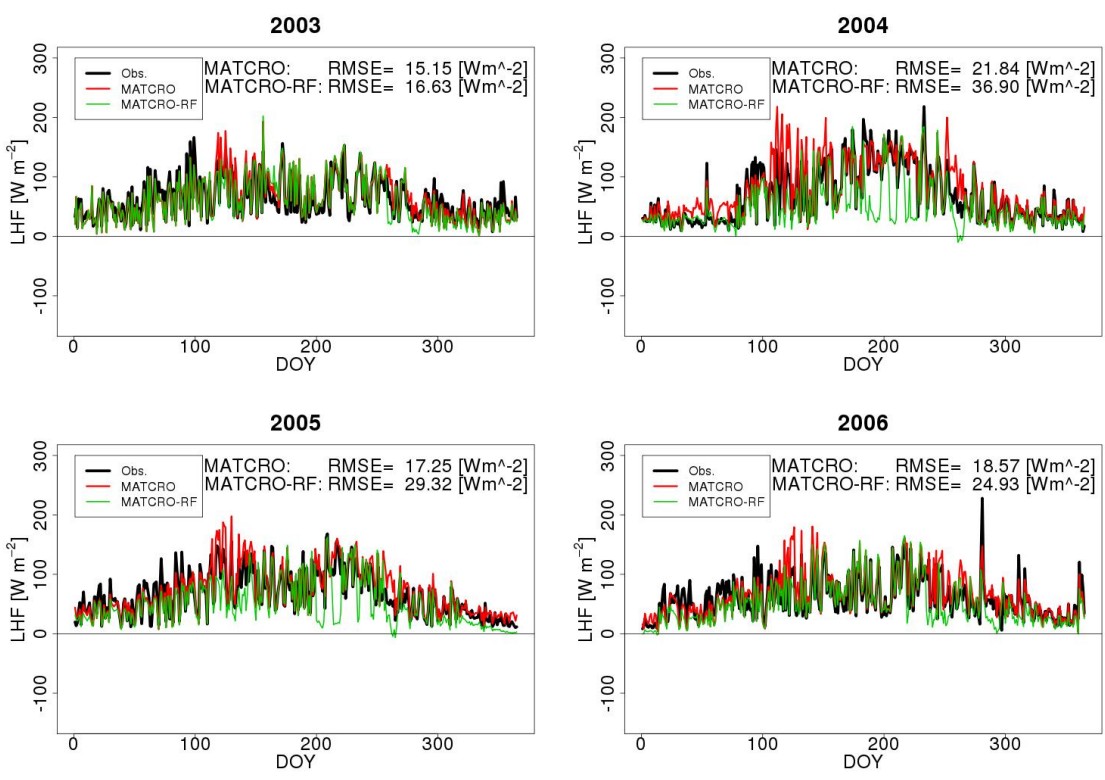

**Figure 15.** Comparison of daily latent heat flux (LHF) between observations and simulations by MATCRO and MATCRO-RF. DOY: The number of days from Jan. 1.

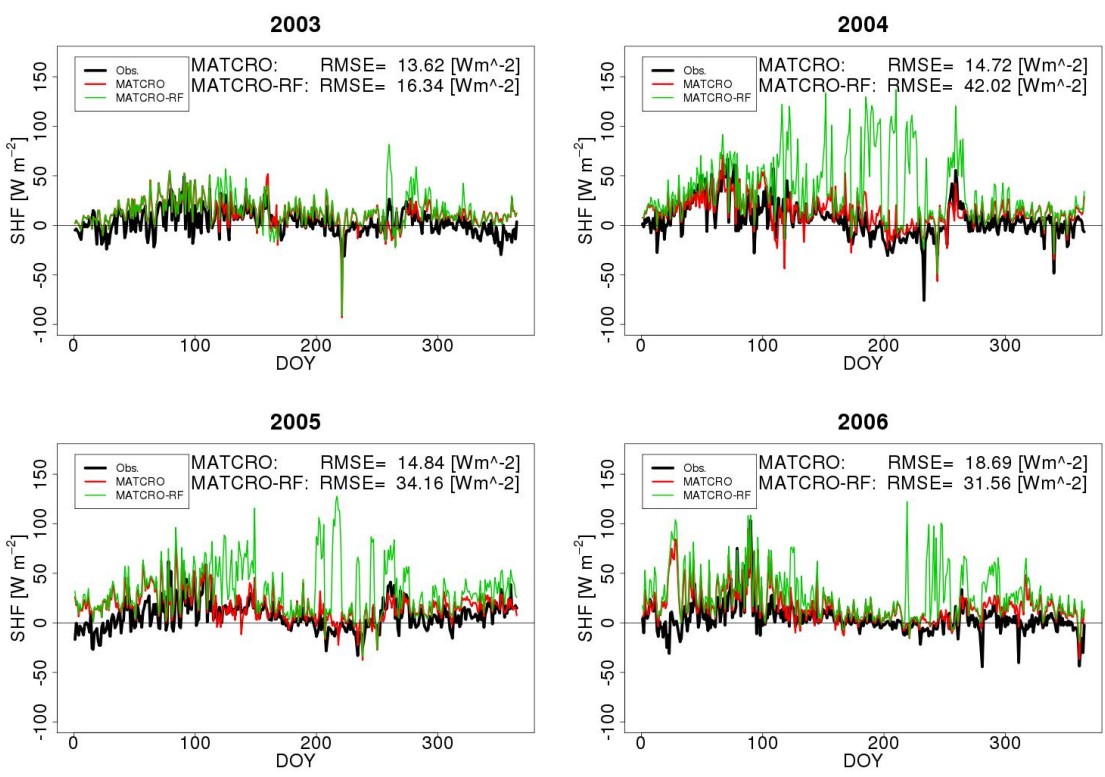

**Figure 16.** Comparison of sensible heat flux (SHF) between observation and simulations by MATCRO and MATCRO-RF. DOY: The number of days from Jan. 1.

**Table 1.** Simulation setting parameters

| Variable | Value | Unit | Description |
|---|---|---|---|
| $C_{\mathrm{a,ppm}}$ | 390 | ppm | atmospheric $CO_2$ concentration |
| $D_{\mathrm{oy,Ie}}$ | - | DOY | DOY of the day that irrigation and flooded surface end |
| $D_{\mathrm{oy,Is}}$ | - | DOY | DOY of the day that irrigation and flooded surface start |
| $D_{\mathrm{oy,sw}}$ | - | DOY | DOY of sowing day |
| $d_{\mathrm{w}}$ | 0.025 | m | depth of surface water |
| $L_{\mathrm{t}}$ | - | degree | latitude of the simulation site |
| $W_{\mathrm{glu,0}}$ | 0.5 | kg ha$^{-1}$ | dry weight of glucose reserve at emergence |
| $W_{\mathrm{lef,0}}$ | 1.0 | kg ha$^{-1}$ | dry weight of leaf at emergence |
| $W_{\mathrm{rot,0}}$ | 1.0 | kg ha$^{-1}$ | dry weight of root at emergence |
| $W_{\mathrm{stm,0}}$ | 1.0 | kg ha$^{-1}$ | dry weight of stem at emergence |
| $z_{\mathrm{a}}$ | 3 | m | reference height at which wind speed is observed |
| $z_{\mathrm{max}}$ | 4 | m | depth of soil layer |
| $z_{\mathrm{t}}$ | 0.05 | m | depth of top soil layer |
| $z_{\mathrm{b}}$ | 2 | m | depth from the soil surface to the upper bound of the bottommost layer of soil |
| $\delta t$ | 1800 | s | time resolution |

**Table 2.** Parameters parameterised

| Variable | Value | Unit | Description |
|---|---|---|---|
| $D_{\text{vs,rot1}}$ | 0.1 | - | 1st point of DVS at which the partition pattern to root changes |
| $D_{\text{vs,rot2}}$ | $D_{\text{vs,h}}$ | - | 2nd point of DVS at which the partition pattern to root changes |
| $D_{\text{vs,lef1}}$ | 0.2 | - | 1st point of DVS at which the partition pattern to leaf changes |
| $D_{\text{vs,lef2}}$ | 0.7 | - | 2nd point of DVS at which the partition pattern to leaf changes |
| $D_{\text{vs,pnc1}}$ | 0.5 | - | 1st point of DVS at which the partition pattern to panicle changes |
| $D_{\text{vs,pnc2}}$ | 0.7 | - | 2nd point of DVS at which the partition pattern to panicle changes |
| $D_{\text{vs,e}}$ | 0.012 | - | DVS at emergence |
| $f_{\text{stc}}$ | 0.288 | - | fraction of glucose allocated to starch reserves |
| $h_{\text{aa}}$ | 0.439 | - | parameter for relationship between LAI and crop height before heading |
| $h_{\text{ab}}$ | 0.675 | - | parameter for relationship between LAI and crop height before heading |
| $h_{\text{ba}}$ | 0.366 | - | parameter for relationship between LAI and crop height after heading |
| $h_{\text{bb}}$ | 0.318 | - | parameter for relationship between LAI and crop height after heading |
| $D_{\text{vs,h}}$ | 0.616 | - | DVS at heading |
| $k_{\text{yld}}$ | 0.675 | - | ratio of crop yield to dry weight of panicle at maturity |
| $k_{\text{S}_{\text{lw}}}$ | 3.5 | - | parameter that represent the relationship between $SLW$ and $DVS$ |
| $G_{\text{ds,m}}$ | 167759940 | K· s | growing degree second at maturity |
| $P_{\text{rot}}$ | 0.25 | - | partition ratio of glucose to root |
| $P_{\text{lef}}$ | 0.5 | - | partition ratio of glucose to leaf from glucose partitioned to shoot |
| $r_{\text{d1,lef}}$ | $5.0 * 10^{-7}$ | $\text{s}^{-1}$ | ratio of leaf death at harvest |
| $S_{\text{lw,mx}}$ | 600 | $\text{kg m}^{-2}$ | maximum specific leaf area |
| $S_{\text{lw,mn}}$ | 350 | $\text{kg m}^{-2}$ | minimum specific leaf area |
| $s_1$ | 0.045 | $\text{K}^{-1}$ | temperature dependence of $\overline{V}_{max,x}$ on $\overline{V}_{m,x}$ |
| $s_2$ | 328 | K | temperature dependence of $\overline{V}_{max,x}$ on $\overline{V}_{m,x}$ |
| $V_{\max}(0)$ | 0.001 | $\text{mol m}^{-2}(l)\,\text{s}^{-1}$ | reference value for maximum Rubisco capacity at the canopy top |
| $D_{\text{vs,tr}}$ | 0.06 | - | DVS at transplanting and at which transplanting shock starts |
| $D_{\text{vs,te}}$ | 0.08 | - | DVS at which transplanting shock ends |

**Table 3.** Observational data used for parameterisation

| Variable | Unit | Description |
|---|---|---|
| $L$ | $m^2(l)\, m^{-2}$ | Leaf area index |
| $D_{oy,tr}$ | day | the number of days of transplanting from Jan. 1 |
| $D_{oy,hd}$ | day | the number of days of heading from Jan. 1 |
| $D_{oy,hv}$ | day | the number of days of harvest from Jan. 1 |
| $h_{gt}$ | m | Crop height |
| $W_{lef}$ | $kg\, ha^{-1}$ | dry matter weight of leaf |
| $W_{stm}$ | $kg\, ha^{-1}$ | dry matter weight of stem |
| $W_{rot}$ | $kg\, ha^{-1}$ | dry matter weight of root |
| $W_{pnc}$ | $kg\, ha^{-1}$ | dry matter weight of panicle |
| $Y_{ld}$ | $kg\, ha^{-1}$ | Yield |

**Table 4.** Observational data used for validation at the parameterisation site

| Variable | Unit | Description |
|---|---|---|
| **Meteorological inputs** | | |
| $P_a$ | Pa | Air pressure |
| $P_r$ | $kg\, m^{-2}\, s^{-1}$ | Precipitation |
| $Q$ | $kg\, kg^{-1}$ | Specific humidity |
| $R_s^d(0)$ | $W\, m^{-2}$ | Downward shortwave radiant flux density at the canopy top |
| $R_l^d(0)$ | $W\, m^{-2}$ | Downward longwave radiant flux density at the canopy top |
| $T_a$ | K | Air temperature |
| $U$ | $m\, s^{-1}$ | Wind speed |
| $D_1^d(0) + S_1^d(0)$ | $W\, m^{-2}$ | Downward radiant flux density for photosynthetic active radiation at the canopy top |
| **Management** | | |
| $d_{wo}$ | m | Observed depth of surface water |
| $D_{oy,tr}$ | DOY | DOY of transplanting day |
| **Outputs** | | |
| $\lambda E$ | $W\, m^{-2}$ | Latent heat flux |
| $H$ | $W\, m^{-2}$ | Sensible heat flux |
| $L$ | - | LAI |
| $T_g$ | - | Surface temperature |
| $W_{sh} + W_{rot}$ | $kg\, ha^{-1}$ | Total biomass |
| $Y_{ld}$ | $kg\, ha^{-1}$ | Yield |

**Table 5.** Soil-type specific parameters

| Variable | Value | Unit | Description | Source |
|---|---|---|---|---|
| $B$ | 5.2 | - | factor for hydraulic conductivity and water potential | Campbell and Norman (1998) |
| $K_{\mathrm{s}}$ | 0.000064 | kg s m$^{-3}$ | hydraulic conductivity at saturation | Campbell and Norman (1998) |
| $w_{\mathrm{sat}}$ | 0.48 | m$^3$ m$^{-3}$ | volumetric concentration of soil water at saturation | Saxton and Rawls (2006) |
| $w_{\mathrm{wlt}}$ | 0.22 | m$^3$ m$^{-3}$ | volumetric concentration of soil water at wilting point | Saxton and Rawls (2006) |
| $\psi_{\mathrm{s}}$ | -2.6 | J kg$^{-1}$ | water potential at saturation | Campbell and Norman (1998) |
| $\rho_{\mathrm{s}}$ | 1390 | kg m$^{-3}$ | bulk density of soil | Saxton and Rawls (2006) |

**Table 6.** RMSEs for daily LHF

| Model | 2003 | 2004 | 2005 | 2006 | Ave. |
|---|---|---|---|---|---|
| MATCRO | 15.15 | 21.84 | 17.25 | 18.57 | 18.20 |
| MATCRO (fixed LAI) | 19.4 | 20.78 | 19.28 | 18.72 | 19.54 |
| MATCRO-RF | 16.63 | 36.90 | 29.32 | 24.93 | 26.95 |

**Table 7.** RMSEs for daily SHF

| Model | 2003 | 2004 | 2005 | 2006 | Ave. |
|---|---|---|---|---|---|
| MATCRO | 13.62 | 14.72 | 14.84 | 18.69 | 15.47 |
| MATCRO (fixed LAI) | 14.69 | 20.30 | 16.93 | 20.68 | 18.15 |
| MATCRO-RF | 16.34 | 42.02 | 34.16 | 31.56 | 31.02 |