# Peer review of "A land surface model combined with a crop growth model for paddy rice (MATCRO-Rice Ver. 1) – Part II: Model validation"

_Geoscientific Model Development, 2016_

## Referee Comment (RC1) · Anonymous Referee #1 · 8 Apr 2016

Masutomi et al. have evaluated 1) part of the water and energy flux terms, 2) net carbon flux and 3) the yield of rice crop at a flux observational site during 2003-2006 using a Crop Growth Model incorporated into a Land Surface Model. As the authors point out representation of interactive crop growth in Land Surface and Earth System Modelling is increasingly important to understand the water, energy, and carbon cycle interactons of climate, especially on regional to local scales.

Overall the article demonstrates that the new model (whose description submitted as a separate article) can reproduce the latent and sensible heat fluxes, biomass growth and rice yield on daily time scale for the 4-yr time period. The coupled Land Surface - Crop Growth model can interactively simulate LAI, crop height and root depth. The

authors imply that the tool can potentially be used to study climate change impact on rice production and crop-climate interactions.

However, I think the authors need to discuss 1) their results in relation to which aspects of the improvements in the new design may be resulting in the better model-observation comparison of each model output, and 2) the validity of the parameters in both wet and dry climatic conditions. The article as such is short and possibly benefit from expanding the evaluation of some of the model output using other available observations. I also feel a comparison to the original MATSIRO simulations wherever possible would certainly help to quantify the improvement in the newly proposed model version.

The article requires revision but the findings would be a step towards advances in land-surface crop modelling if discussed sufficiently.

Specific Comments

1. If the authors mean to validate the new model, the paper should include comparison of LHF and SHF (Fig. 7 & 8) to the original simulations of the parent LSM (without the present modifications or interactive crop growth and development). How different are the stomatal conductance and the moisture and temperature of the soil column in the parent LSM when uncoupled to the CGM?

2. I wonder whether the authors have compared the water and energy flux terms on shorter time scales than daily, say, to look at the diurnal variations of LHF and SHF in both obs and simulations during the various stages of the crop growth. It would be useful to understand the impact of crop-climate interactions on the water balance on sub-daily timescales, which is an ongoing challenge in climate modelling.

Section 1, L19: Expand MATCRO.

Section 2, L16-L17: Why only 2003-2006 was chosen instead of 2001-2006 (when the observations seem to be available according to the given website)? Justify here.

Section 2.2, L1-10: Equations to calculate the soil state are missing here.

Section 3, L25-L26 & Section 4, L29: Either remove the last sentences or explain shortly how.

Section 5, L22: Add here a couple of sentences on what changes in parameters/processes in the model may have resulted in the important feature of the model using a schematic of the processes represented in each module of the coupled model.
* * *

---

## Referee Comment (RC2) · C. Müller (Referee) · 18 Apr 2016

Masutomi and co-authors present a paper on the validation of a new coupling of an existing land-surface model MATSIRO and a crop growth model for rice, which seems to be a new development based on existing modeling approaches. With this model development, the authors aim to support model studies on the effects of agricultural land use on climate as well as hydrology. Masutomi and co-authors are right in their assessment of the need and applicability of agricultural modules in climate and hydrology studies and a new modeling approach with a focus on rice systems is thus highly welcome.

I have, however, substantial concerns with respect to the validity of this "validation paper". First of all, I'm expecting that the MATCRO-Rice model is going to be applied at river catchment or continental to global scale. Even though the envisioned scale of future application is not explicitly mentioned, I'm assuming so, as the model development is motivated by the wish to better study the effects of agricultural production systems (here: rice production) on hydrology and climate. Yet, the validation only provides a comparison to a single site, which even lacks a central data element (crop yield) that had to be deduced by trend extrapolation. The model validation presented then turns out to be a demonstration of model calibration to a single site and then can reproduce much of the observed dynamics at this site. Yet, it remains unclear how the model would perform at sites where such intensive calibration is not possible for the lack of data. The authors seem to constrain their "validation" to this one site as it seems to be the only eddy flux measurement site for rice production systems, but clearly that is no proof of model skill. As suggested by the other reviewer, I would much appreciate if the improvements of the MATCRO-Rice model could be evaluated against the original MATSIRO simulations and if there was an evaluation of model skill apart from the calibration site. Authors are advised to consult e.g. Luo et al. (2012) for possible data sets and metrics and Iizumi et al. (2014) for yield data.

There seems to be a misconception on the net carbon flux between land and atmosphere. The authors claim that the models ability to reproduce the biomass accumulation is an indicator for its applicability to simulate the net carbon flux: "As indicated by the figure, the simulated total biomass was in good agreement with the observations. Hence, we conclude that the model has high accuracy for simulating net carbon flux during growing period" (page 5, similar on page 6). Yet, the net carbon flux is composed of net carbon uptake by plants (NPP) and the mineralization of soil organic matter as well as other disturbances such as fire, pest outbreaks etc. which may not be too relevant here. But the soil respiration flux is a central aspect in this and cannot be ignored.

Minor remark: Page 10: DVS is likely "development stage" not "dynamic vegetation

model"?

The paper as presented here addresses an important in land surface and agricultural modeling but fails to validate the model or to evaluate the model performance. Model performance needs to be evaluated against independent data sources, the improvement compared to the original model should be quantified (and eventually assessed against possible shortcomings) and the validation/evaluation should be performed at the scale of envisioned application, not just at a single point. Also, the model evaluation should address the ability to reproduce spatial and/or temporal patterns.

Unless substantially extended to justify a publication on its own, this paper could well be merged into the model description paper at http://www.geosci-model-dev-discuss.net/gmd-2016-28/, as also suggested by a reviewer there.

References

* Iizumi T, Yokozawa M, Sakurai G, Travasso M I, Romanenkov V, Oettli P, Newby T, Ishigooka Y and Furuya J 2014 Historical changes in global yields: major cereal and legume crops from 1982 to 2006 Global Ecology and Biogeography 23 346-57

* Luo Y Q, Randerson J T, Abramowitz G, Bacour C, Blyth E, Carvalhais N, Ciais P, Dalmonech D, Fisher J B, Fisher R, et al. 2012 A framework for benchmarking land models Biogeosciences 9 3857-74

---

## Author Comment (AC1) · 19 May 2016

**Response to Anonymous Referee#1 (gmd-2016-29)**

###################################################################

*However, I think the authors need to discuss 1) their results in relation to which aspects of the improvements in the new design may be resulting in the better model-observation comparison of each model output, and 2) the validity of the parameters in both wet and dry climatic conditions. The article as such is short and possibly benefit from expanding the evaluation of some of the model output using other available observations. I also feel a comparison to the original MATSIRO simulations wherever possible would certainly help to quantify the improvement in the newly proposed model version.*

###################################################################

According to your and other referees' comments, we will add the results of two types of simulations into the revised manuscript: the effects of modifications from the original model and the validation of the model at the sites which are independent from the parameterization site.

The parameters given in the manuscript are not valid under dry climatic conditions, because the current version of the model is designed for irrigated rice. This is one major limitation of the current version of MATCRO-Rice. We have recognized the limitation and described it in the concluding remarks, P20 L23-24. The limitation will be addressed in the future paper, because it is out of the scope of the present study that focus on irrigated rice.

###################################################################

*If the authors mean to validate the new model, the paper should include comparison of LHF and SHF (Fig. 7 & 8) to the original simulations of the parent LSM (without the present modifications or interactive crop growth and development). How different are the stomatal conductance and the moisture and temperature of the soil column in the parent LSM when uncoupled to the CGM?*

###################################################################

As we mentioned above, we will add the results of the simulations to evaluate the effects of modifications from the original model.

####################################################################
*I wonder whether the authors have compared the water and energy flux terms on shorter time scales than daily, say, to look at the diurnal variations of LHF and SHF in both obs and simulations during the various stages of the crop growth. It would be useful to understand the impact of crop-climate interactions on the water balance on sub-daily timescales, which is an ongoing challenge in climate modelling.*
####################################################################
We fully agree with your opinion and suggestion. We will add the results of the comparisons of hourly LHF and SHF between observations and simulations.

####################################################################
*Section 1, L19: Expand MATCRO.*
####################################################################
We will add brief explanation on the model and its name.

####################################################################
*Section 2, L16-L17: Why only 2003-2006 was chosen instead of 2001-2006 (when the observations seem to be available according to the given website)? Justify here.*
####################################################################
The flux tower was moved in the paddy field in April 2003. Thereafter the obtained flux data have been more representative of the field, where the rice sampling was conducted. We will add the above explanation in the revised manuscript.

####################################################################
*Section 2.2, L1-10: Equations to calculate the soil state are missing here.*

\#\#\#\#\#\#\#\#\#\#\#\#\#\#\#\#\#\#\#\#\#\#\#\#\#\#\#\#\#\#\#\#\#\#\#\#\#\#\#\#\#\#\#\#\#\#\#\#\#\#\#\#\#\#\#\#\#\#\#\#\#\#\#\#\#\#\#\#

All the equations are shown in the model description paper. In our understanding of your comment, you pointed out that the discrete equations to calculate the soil state should be described in the manuscript. Therefore, we will add the discrete equations to calculate the soil state.

\#\#\#\#\#\#\#\#\#\#\#\#\#\#\#\#\#\#\#\#\#\#\#\#\#\#\#\#\#\#\#\#\#\#\#\#\#\#\#\#\#\#\#\#\#\#\#\#\#\#\#\#\#\#\#\#\#\#\#\#\#\#\#\#\#\#\#\#\#

*Section 3, L25-L26 & Section 4, L29: Either remove the last sentences or explain shortly how.*

\#\#\#\#\#\#\#\#\#\#\#\#\#\#\#\#\#\#\#\#\#\#\#\#\#\#\#\#\#\#\#\#\#\#\#\#\#\#\#\#\#\#\#\#\#\#\#\#\#\#\#\#\#\#\#\#\#\#\#\#\#\#\#\#\#\#\#\#\#

We will remove the sentences you pointed out.

\#\#\#\#\#\#\#\#\#\#\#\#\#\#\#\#\#\#\#\#\#\#\#\#\#\#\#\#\#\#\#\#\#\#\#\#\#\#\#\#\#\#\#\#\#\#\#\#\#\#\#\#\#\#\#\#\#\#\#\#\#\#\#\#\#\#\#\#\#

*Section 5, L22: Add here a couple of sentences on what changes in parameters/processes in the model may have resulted in the important feature of the model using a schematic of the processes represented in each module of the coupled model.*

\#\#\#\#\#\#\#\#\#\#\#\#\#\#\#\#\#\#\#\#\#\#\#\#\#\#\#\#\#\#\#\#\#\#\#\#\#\#\#\#\#\#\#\#\#\#\#\#\#\#\#\#\#\#\#\#\#\#\#\#\#\#\#\#\#\#\#\#\#

In the concluding remarks of the revised manuscript, we will discuss the important feature of the coupled model, referring the results of the new simulations to evaluate the effects of modifications from the original model.

---

## Author Response (AR1)

**Response to Anonymous Referee#1 (gmd-2016-29)**

######################################################################
*However, I think the authors need to discuss 1) their results in relation to which aspects of the improvements in the new design may be resulting in the better model-observation comparison of each model output, and 2) the validity of the parameters in both wet and dry climatic conditions. The article as such is short and possibly benefit from expanding the evaluation of some of the model output using other available observations. I also feel a comparison to the original MATSIRO simulations wherever possible would certainly help to quantify the improvement in the newly proposed model version.*
######################################################################
According to your and other referees' comments, we added the results of two types of simulations into the revised manuscript: the effects of modifications from the original model (Section 5) and the validation of the model at the sites which are independent from the parameterization site (Section 4.2). Along with the modification of the manuscript, we changed the structure of the manuscript (Section 2: Numerical setting and method; Section 3: Parameterisation; Section 4: Validation; 5: Effects of modification).

######################################################################
*If the authors mean to validate the new model, the paper should include comparison of LHF and SHF (Fig. 7 & 8) to the original simulations of the parent LSM (without the present modifications or interactive crop growth and development). How different are the stomatal conductance and the moisture and temperature of the soil column in the parent LSM when uncoupled to the CGM?*
######################################################################
As we mentioned above, we added the results of the simulations to evaluate the effects of modifications of the original model (Section 5).

##################################################################

*I wonder whether the authors have compared the water and energy flux terms on shorter time scales than daily, say, to look at the diurnal variations of LHF and SHF in both obs and simulations during the various stages of the crop growth. It would be useful to understand the impact of crop-climate interactions on the water balance on sub-daily timescales, which is an ongoing challenge in climate modelling.*

##################################################################

We fully agree with your opinion and suggestion. We added the results of the comparisons of half hourly LHF and SHF between observations and simulations (Figures 8 and 10).

##################################################################

*Section 1, L19: Expand MATCRO.*

##################################################################

We added brief explanation on the model and its name (P2, L1-2).

##################################################################

*Section 2, L16-L17: Why only 2003-2006 was chosen instead of 2001-2006 (when the observations seem to be available according to the given website)? Justify here.*

##################################################################

The flux tower was moved in the paddy field in April 2003. Thereafter the obtained flux data have been more representative of the field, where the rice sampling was conducted. We added the above explanation in the revised manuscript (P5, L22-24).

##################################################################

*Section 2.2, L1-10: Equations to calculate the soil state are missing here.*

##################################################################

All the equations are shown in the model description paper. Therefore we referred to the sections where the equations are described in the model description paper (P2, L18; P2, L28-31).

###############################################################
*Section 3, L25-L26 & Section 4, L29: Either remove the last sentences or explain shortly how.*
###############################################################
We removed the sentences you pointed out.

###############################################################
*Section 5, L22: Add here a couple of sentences on what changes in parameters/processes in the model may have resulted in the important feature of the model using a schematic of the processes represented in each module of the coupled model.*
###############################################################
In the concluding remarks of the revised manuscript, we discussed the important feature of the coupled model, referring the results of the new simulations to evaluate the effects of modifications from the original model (P9, L5-8).

**Response to C. Muller (gmd-2016-29)**

###################################################################

*I have, however, substantial concerns with respect to the validity of this "validation paper". First of all, I'm expecting that the MATCRO-Rice model is going to be applied at river catchment or continental to global scale. Even though the envisioned scale of future application is not explicitly mentioned, I'm assuming so, as the model development is motivated by the wish to better study the effects of agricultural production systems (here: rice production) on hydrology and climate. Yet, the validation only provides a comparison to a single site, which even lacks a central data element (crop yield) that had to be deduced by trend extrapolation. The model validation presented then turns out to be a demonstration of model calibration to a single site and then can reproduce much of the observed dynamics at this site. Yet, it remains unclear how the model would perform at sites where such intensive calibration is not possible for the lack of data. The authors seem to constrain their "validation" to this one site as it seems to be the only eddy flux measurement site for rice production systems, but clearly that is no proof of model skill. As suggested by the other reviewer, I would much appreciate if the improvements of the MATCRO-Rice model could be evaluated against the original MATSIRO simulations and if there was an evaluation of model skill apart from the calibration site. Authors are advised to consult e.g. Luo et al. (2012) for possible data sets and metrics and Iizumi et al. (2014) for yield data.*

###################################################################

According to your and other referees' comments, we added the results of two types of simulations into the revised manuscript: the effects of modifications of the original model (Section 5) and the validation of the model at the sites which are independent from the parameterization site (Section 4.2). Along with the modification of the manuscript, we changed the structure of the manuscript (Section 2: Numerical setting and method; Section 3: Parameterisation; Section 4: Validation; 5: Effects of modification).

###################################################################

*There seems to be a misconception on the net carbon flux between land and atmosphere. The*

*authors claim that the models ability to reproduce the biomass accumulation is an indicator for its applicability to simulate the net carbon flux: "As indicated by the figure, the simulated total biomass was in good agreement with the observations. Hence, we conclude that the model has high accuracy for simulating net carbon flux during growing period" (page 5, similar on page 6). Yet, the net carbon flux is composed of net carbon uptake by plants (NPP) and the mineralization of soil organic matter as well as other disturbances such as fire, pest outbreaks etc. which may not be too relevant here. But the soil respiration flux is a central aspect in this and cannot be ignored.*

###############################################################

Your comments are exactly right. We simulated just net carbon uptake by plants, which is one element of net carbon flux between land and the atmosphere. We replaced all phrases "net carbon flux" by "net carbon uptake by crop" in the revised manuscript.

###############################################################

*Page 10: DVS is likely "development stage" not "dynamic vegetation model"?*

###############################################################

That's is our mistake. We modified them in the revised manuscript.

###############################################################

*The paper as presented here addresses an important in land surface and agricultural modeling but fails to validate the model or to evaluate the model performance. Model performance needs to be evaluated against independent data sources, the improvement compared to the original model should be quantified (and eventually assessed against possible shortcomings) and the validation/evaluation should be performed at the scale of envisioned application, not just at a single point. Also, the model evaluation should address the ability to reproduce spatial and/or temporal patterns.*

*Unless substantially extended to justify a publication on its own, this paper could well be merged into the model description paper at http://www.geosci-model-dev- discuss.net/gmd-2016-28/, as also suggested by a reviewer there.*

###############################################################

The revised manuscript was substantially extended by adding two types of the simulations: the evaluation of the model using the independent data and the comparison of the model with the original model. In addition, the journal guideline of GMD admits the separate submission of the model description and evaluation papers, if the evaluation is extensive. Therefore, we separately submitted the revised manuscript of the model description and evaluation papers.

[revised manuscript text omitted]
_{\text{Eg}}$, $C_{\text{Ec}}$, $C_{\text{Hg}}$, $C_{\text{Hc}}$, $C_{\text{M}}$, and $C_{\text{Mg}}$; Section 3.3 in Masutomi et al. (2016)), and variables related to carbon assimilation ( $\overline{A}_{\text{n},x}, c_{\text{i},x},$ and $\overline{g}_{\text{st},x}$; Section 4.1 in Masutomi et al. (2016)) .

We set  $z_{\text{a}}$ =  3. $CO_2$ concentration ($C_{\text{a,ppm}}$) and the depth of surface water ($d_{\text{w}}$) were set at 390 ppm and 0.025 m, respectively. The initial dry weight of each organ was set at 1 kg ha$^{-1}$ for leaf ($W_{\text{lef},0}$), stem ($W_{\text{stm},0}$), and root ($W_{\text{rot},0}$) and at 0.5 kg ha$^{-1}$ for glucose reserve in leaf ($W_{\text{glu},0}$).

$D_{\text{ov,le}}$, $D_{\text{ov,ls}}$, $D_{\text{ov,sw}}$, and $L_{\text{t}}$ depend on the simulations. Values for these parameters are shown in the sections of each simulation.

**3 Parameterisation**

**3.1**

      Table 2 shows model parameters parameterised in the present paper. All parameters are parameterised using observations, the literature, and assumptions. The method of the parameterisation is explained in this section.

**3.1 Parameterisation site and observation data**

Table 3 shows the observational data used for parameterisation. The data were observed from 2003 to 2006 at a site which is located in Tsukuba, Japan (Lat: 36° 03' 14.3" N; Lon: 140° 01' 36.9" E), at 13 m above sea level. The climatic zone of the site is temperate, with the mean annual air temperature 13.7°C and precipitation 1200 mm. The soil type is clay loam. The variety planted at the site is "Koshihikari", which is the most planted variety in Japan.

Biomass for each organ ($W_{\mathrm{lef}}$, $W_{\mathrm{pnc}}$, $W_{\mathrm{rot}}$, and $W_{\mathrm{stm}}$) and leaf area index ($L$) were measured nearly every two weeks. At each measuring time, ten stands were sampled from the fields. Yield ($Y_{\mathrm{ld}}$) and phenological dates including transplanting ($D_{\mathrm{oy,tr}}$), heading ($D_{\mathrm{oy,hd}}$), and harvest ($D_{\mathrm{oy,hv}}$) were observed every year. The values of observed yield are the husked rice yield with 15% water content. The rice grains for measuring yield were sampled from the whole fields of the    observational site. The crop height ($h_{\mathrm{gt}}$) was measured on average every 5 days.

**3.2 Phenology**

Phenological parameters that represent development stages ($D_{\mathrm{vs,e}}$, $D_{\mathrm{vs,h}}$, $G_{\mathrm{ds,m}}$, $D_{\mathrm{vs,tr}}$, and $D_{\mathrm{vs,te}}$) were parameterised. First, we calculated $D_{\mathrm{vs}}$s at heading and $G_{\mathrm{ds,m}}$s from 2003 to 2006 using the phenological model given by Masutomi et al. (2016). The mean values were set to $D_{\mathrm{vs,h}}$ and $G_{\mathrm{ds,m}}$, resulting in $D_{\mathrm{vs,h}} = 0.616$ and $G_{\mathrm{
[revised manuscript text omitted]

[Figure]

**Figure 5.** Relationship between specific leaf weight and  development stage (DVS)

[Figure]

**Figure 6.** Rubisco-limited photosynthesis rate

[Figure]

**Figure 7.** Comparison of daily latent heat flux (LHF) between simulations and observations. DOY: The number of days from Jan. 1.

[Figure]

**Figure 8.** Comparison of half-hourly latent heat flux (LHF) between simulations and observations.

[Figure]

**Figure 9.** Comparison of sensible heat flux (SHF) between simulations and observations. DOY: The number of days from Jan. 1.

[Figure]

**Figure 10.** Comparison of half-hourly sensible heat flux (SHF) between simulations and observations.

[Figure]

**Figure 11.** Comparison of total biomass between simulations and observations during growing periods from 2003 to 2006. Circles indicate mean values of observations and the ranges indicate standard deviation of observations. Red lines denotes simulations. DOY: The number of days from Jan. 1.

[Figure]

**Figure 12.** Comparison of yields between simulations and observations

[Figure]

**Figure 13.** Comparison of yields over Japan between simulations and observations

[Figure]

**Figure 14.** Comparison of LAI between observations, simulations by MATCRO, and the default value of MATSIRO

[Figure]

**Figure 15.** Comparison of daily latent heat flux (LHF) between observations and simulations by MATCRO and MATCRO-RF. DOY: The number of days from Jan. 1.

[Figure]

**Figure 16.** Comparison of sensible heat flux (SHF) between observation and simulations by MATCRO and MATCRO-RF. DOY: The number of days from Jan. 1.

[Figure]

**Figure 17.** Surface temperature between observation and simulations by MATCRO and MATCRO-RF. DOY: The number of days from Jan. 1.

**Table 1.**  Simulation setting parameters

| Variable | Value |
|---|---|
|  $C_{a,ppm}$ | 390 |
|  $D_{oy,le}$ |  |
|  $D_{oy,ls}$ |  |
|  $D_{oy,sw}$ |  |
|  $d_w$ |  |
|  $L_t$ |  |
|  $W_{glu,0}$ |  $W_{glu,0}$ 0.5 |
|  $W_{lef,0}$ | 1.0 |
|  $W_{rot,0}$ | 1.0 |
|  $W_{stm,0}$ | 1.0 |
|  $z_a$ | 3 |
|  $z_{max}$ | 4 |
|  $z_t$ | 0.05 |
|  $z_b$ | 2 |
| $\delta t$ | 1800 |

**Table 2.** Parameters parameterised

| Variable | Value | Unit | Description |
|---|---|---|---|
| $D_{vs,rot1}$ | 0.1 | - | 1st point of DVS at which the partition pattern to root changes |
| $D_{vs,rot2}$ | $D_{vs,h}$ | - | 2nd point of DVS at which the partition pattern to root changes |
| $D_{vs,lef1}$ | 0.2 | - | 1st point of DVS at which the partition pattern to leaf changes |
| $D_{vs,lef2}$ | 0.7 | - | 2nd point of DVS at which the partition pattern to leaf changes |
| $D_{vs,pnc1}$ | 0.5 | - | 1st point of DVS at which the partition pattern to panicle changes |
| $D_{vs,pnc2}$ | 0.7 | - | 2nd point of DVS at which the partition pattern to panicle changes |
| $D_{vs,e}$ | 0.012 | - | DVS at emergence |
| $f_{stc}$ | 0.288 | - | fraction of glucose allocated to starch reserves |
| $h_{aa}$ | 0.439 | - | parameter for relationship between LAI and crop height before heading |
| $h_{ab}$ | 0.675 | - | parameter for relationship between LAI and crop height before heading |
| $h_{ba}$ | 0.366 | - | parameter for relationship between LAI and crop height after heading |
| $h_{bb}$ | 0.318 | - | parameter for relationship between LAI and crop height after heading |
| $D_{vs,h}$ | 0.616 | - | DVS at heading |
| $k_{yld}$ | 0.675 | - | ratio of crop yield to dry weight of panicle at maturity |
| $k_{S_{lw}}$ | 3.5 | - | parameter that represent the relationship between $SLW$ and $DVS$ |
| $G_{ds,m}$ | 167759940 | K· s | growing degree second at maturity |
| $P_{rot}$ | 0.25 | - | partition ratio of glucose to root |
| $P_{lef}$ | 0.5 | - | partition ratio of glucose to leaf from glucose partitioned to shoot |
| $r_{d1,lef}$ | $5.0 * 10^{-7}$ | $s^{-1}$ | ratio of leaf death at harvest |
| $S_{lw,mx}$ | 600 | kg m$^{-2}$ | maximum specific leaf area |
| $S_{lw,mn}$ | 350 | kg m$^{-2}$ | minimum specific leaf area |
| $s_1$ | 0.045 | K$^{-1}$ | temperature dependence of $\overline{V}_{max,x}$ on $\overline{V}_{m,x}$ |
| $s_2$ | 328 | K | temperature dependence of $\overline{V}_{max,x}$ on $\overline{V}_{m,x}$ |
| $V_{max}(0)$ | 0.001 | mol m$^{-2}(l)$ s$^{-1}$ | reference value for maximum Rubisco capacity at the canopy top |
| $D_{vs,tr}$ | 0.06 | - | DVS at transplanting and at which transplanting shock starts |
| $D_{vs,te}$ | 0.08 | - | DVS at which transplanting shock ends |

**Table 3.** Observational data used for parameterisation

| Variable | Unit | Description |
|---|---|---|
| $L$ | $\mathrm{m^2}(l)\,\mathrm{m^{-2}}$ | Leaf area index |
| $D_{\mathrm{oy,tr}}$ | day | the number of days of transplanting from Jan. 1 |
| $D_{\mathrm{oy,hd}}$ | day | the number of days of heading from Jan. 1 |
| $D_{\mathrm{oy,hv}}$ | day | the number of days of harvest from Jan. 1 |
| $h_{\mathrm{gt}}$ | m | Crop height |
| $W_{\mathrm{lef}}$ | $\mathrm{kg\,ha^{-1}}$ | dry matter weight of leaf |
| $W_{\mathrm{stm}}$ | $\mathrm{kg\,ha^{-1}}$ | dry matter weight of stem |
| $W_{\mathrm{rot}}$ | $\mathrm{kg\,ha^{-1}}$ | dry matter weight of root |
| $W_{\mathrm{pnc}}$ | $\mathrm{kg\,ha^{-1}}$ | dry matter weight of panicle |
| $Y_{\mathrm{ld}}$ | $\mathrm{kg\,ha^{-1}}$ | Yield |

**Table 4.** Observational data used for validation at the parameterisation site

| Variable | Unit | Description |
|---|---|---|
| **Meteorological inputs** | | |
| $P_{\mathrm{a}}$ | Pa | Air pressure |
| $P_{\mathrm{r}}$ | $\mathrm{kg\,m^{-2}\,s^{-1}}$ | Precipitation |
| $Q$ | $\mathrm{kg\,kg^{-1}}$ | Specific humidity |
| $R_{\mathrm{s}}^{\mathrm{d}}(0)$ | $\mathrm{W\,m^{-2}}$ | Downward shortwave radiant flux density at the canopy top |
| $R_{\mathrm{l}}^{\mathrm{d}}(0)$ | $\mathrm{W\,m^{-2}}$ | Downward longwave radiant flux density at the canopy top |
| $T_{\mathrm{a}}$ | K | Air temperature |
| $U$ | $\mathrm{m\,s^{-1}}$ | Wind speed |
| $D_{\mathrm{l}}^{\mathrm{d}}(0)+S_{\mathrm{l}}^{\mathrm{d}}(0)$ | $\mathrm{W\,m^{-2}}$ | Downward radiant flux density for photosynthesis active radiation at the canopy top |
| **Management** | | |
| $d_{\mathrm{wo}}$ | m | Observed depth of surface water |
| $D_{\mathrm{oy,tr}}$ | DOY | DOY of transplanting day |
| **Outputs** | | |
| $\lambda E$ | $\mathrm{W\,m^{-2}}$ | Latent heat flux |
| $H$ | $\mathrm{W\,m^{-2}}$ | Sensible heat flux |
| $L$ | - | LAI |
| $T_{\mathrm{g}}$ | - | Surface temperature |
| $W_{\mathrm{sh}}+W_{\mathrm{rot}}$ | $\mathrm{kg\,ha^{-1}}$ | Total biomass |
| $Y_{\mathrm{ld}}$ | $\mathrm{kg\,ha^{-1}}$ | Yield |

**Table 5.** Soil-type specific parameters

| Variable | Value | Unit | Description | Source |
|----------|-------|------|-------------|--------|
| $B$ | 5.2 | - | factor for hydraulic conductivity and water potential | Campbell and Norman (1998) |
| $K_s$ | 0.000064 | $\text{kg s m}^{-3}$ | hydraulic conductivity at saturation | Campbell and Norman (1998) |
| $w_{\text{sat}}$ | 0.48 | $\text{m}^3 \text{m}^{-3}$ | volumetric concentration of soil water at saturation | Saxton and Rawls (2006) |
| $w_{\text{wlt}}$ | 0.22 | $\text{m}^3 \text{m}^{-3}$ | volumetric concentration of soil water at wilting point | Saxton and Rawls (2006) |
| $\psi_s$ | -2.6 | $\text{J kg}^{-1}$ | water potential at saturation | Campbell and Norman (1998) |
| $\rho_s$ | 1390 | $\text{kg m}^{-3}$ | bulk density of soil | Saxton and Rawls (2006) |

---

## Author Response (AR2)

**Response to Editor's comments**

####################################################################

*The reviewers seemed to have satisfied with the modification especially in terms of the added information. However, the insufficient description on the simulation and validation specifications reduces the value of this paper still. Especially, the comparison of yield over Japan does not see much meanings without proper justification for the difference in absolute number between SIM and OBS, and without proper discussion on their reason why.*

####################################################################

The error was caused by the following three points: (1) bias in shortwave radiation; (2) the large default value for the ratio of PAR to shortwave radiation; (3) our mistake in input data handling. After rectifying the three points, we recalculated rice yields over Japan. The revised result shows the error between simulations and observation is not large (Figure 13). However, the model still tends to overestimate the observations. One of the reasons for the simulation errors is thought to be the method of parameterisation. For the simulations over Japan, we used parameters parameterised for a variety "Koshihikari" only at one site in Japan (Section 3), although there is a large diversity in agricultural management and technique, and rice varieties planted throughout Japan. Therefore, parameterisation at a large scale is necessary for better large scale simulations. We described the method for rectifying the points (1) and (2) (P7, L9-13), and added the discussion on the error to the revised manuscript (P7, L24-27).

####################################################################

*As the Reviewer 1 suggested, the merging with the companion (part I) paper is another good idea to make the effectiveness on understanding the performance of*

*MATCRO-Rice model better.*

*###############################################################*

The companion paper has been accepted. We think separate but simultaneous publication is a good option. We have the publisher wait for publishing the companion paper.

*###############################################################*

*Overall, I require the authors to revise again according to the Reviewer's comments.*

*###############################################################*

According to the reviewers' comments, we revised the manuscript. Each response to their comments is described below.

**Response to Referee #2's comments**

*The authors have made some effort to actually supply an evaluation of model performance against independent data, which I acknowledge. Generally the paper reads fine. I would suggest extracting much of the numeric information that is given in the text (e.g. RMSE values for different model versions for 4 years and the 4-year mean or parameter values) into one table which would make a comparison across settings much easier.*

################################################################

We added Tables 6 and 7 showing the RMSEs of daily LHF and SHF for three different model versions.

################################################################

*The assessment of yields lacks some important detail on how simulated crop yields were aggregated (average without area weight?) and a simple average across grid-cells with any part of Japan in it does not seem to be the right approach to aggregate simulated yields. This should be clarified and – if needed – rectified.*

################################################################

In the previous manuscript, we used a simple average across grids. As you pointed, it is not a good way for calculating average yields. Therefore, we changed the method for calculating national average yields. In the revised method, rice yields for all prefectures in Japan were simulated from 1991 to 2010, and then the average national rice yield was calculated for each year from the prefectural yields weighted by the prefectural planting areas. In addition, the observed sowing and harvesting dates for each prefecture are used in the yield simulations. The method described here was added to the revised manuscript (Section 4.2.1).

################################################################

*The explanation of incorrect simulated surface temperatures under omission of flooded areas as the main reason why MATCRO works better than MATSIRO does not hold for 2003, where the MATCRO-RF version is closer to observed surface T than the MATCRO version, whereas SHF and LHF fluxes are simulated better by MATCRO than MATCRO-RF in all cases. So the underlying mechanism seems to be a bit more complicated than that and should be discussed in that way.*

################################################################

We agree with your opinion. It is not reasonable to say that the errors in surface temperature caused the large errors of LHF and SHF. In the revised manuscript, we deleted the figure of surface temperature, and described simple but reasonable discussions on the errors (P8, L23-27).

################################################################

*In several places units are missing or parameters are listed within the text where it's not clear to which equations these would actually refer to. If all these equations are in the companion paper, could readers be referred to these?*

################################################################

Some missing units are added to the revised manuscript ("seconds" at P2, L17; "m" at P3, L1; "Ks" at P3, L22).

We think that separate but simultaneous publication with the companion paper will help readers.

################################################################

*I still think that this paper could well be merged with the companion paper, but if the editor and authors see this differently I'm happy to concur.*

######################################################################

We understand your concern. However, the companion paper has been accepted. We think separate but simultaneous publication is a good option. We have the publisher wait for publishing the companion paper.

Detailed comments:

######################################################################

*P1, L1: two types of validation*

######################################################################

We modified the mistake (P1, L1; P5, L4).

######################################################################

*P3, L3: please provide units. I suppose it's 1800 seconds?*

######################################################################

Yes, it is. We added "seconds" (P2, L17).

######################################################################

*P5, L5: I suggest saying "biomass partitioning" throughout as not only glucose is partitioned*

######################################################################

MATCRO partitions carbohydrates in leaves, in the form of glucose, into each organ, according to the MACROS (Penning de Vries et al., 1989). Therefore, we do not think that the change which you suggested is a good option. However, we added a sentence on glucose partitioning (P3, L29-30).

###############################################################
* P6, L24 "photosynthetic active radiation"
###############################################################

We corrected it (P5, L13; in Table 4).

###############################################################
* P8, L24: what kind of averaging was used? I would expect that there is some area weighting with rice harvested area per grid cell, but as there is no mentioning of any land-use input data, this does not seem to be the case? National crop yield statistics certainly reflect the inhomogeneous distribution of cropland within the country and therefore the aggregation of simulated data should as well.
###############################################################

As described above, we changed the method for averaging and added the explanation on the method to the revised manuscript (Section 4.2.1).

###############################################################
* P8, L30 "can reproduce variability correctly" is possibly stretching it a bit… maybe rephrase to something like "can substantial shares of weather-induced variability" (even though you don't know how much of the observed variability is actually weather-driven).
###############################################################

According to your suggestion, we modified the sentence (P7, L 23-24).

###############################################################
* P9, L13: This is not an equation but a parameter? For what equation?

################################################################
In the original LSM, MATSIRO, the maximum amount of water that canopy can hold depends on LAI. The equation showed it. But it was confusing. Therefore, we simplified the sentence in order to make it easier to understand (P8, L10-11).

################################################################
*P9, L29: "correctly" seems to overrate the skill here. MATCRO-RF is "more correct" in 2003?*
################################################################
As we mentioned above, we modified the discussion (P8, L23-27).

################################################################
*Figures 15 and 16: The RMSE values given refer to MATCRO or MATCRO-RF? Why don't you supply both as in Fig 17?*
################################################################
We added the information in Figures 15 and 16.

####################################################################

*The authors have modified a lot according to the reviewers' comments. Most of them satisfied me. However, the simulation over Japan islands does not make any sense for validating the MATCRO-Rice. The long latitudinal distance of Japan cannot guarantee that a single simulation can depict the averaged behavior of bio-geographical distribution of rice-plant responses to interannual climatic change. I recommend deleting this part. Otherwise, the authors should mention about the detail of input data and model simulation specification, also have to discuss the reason why the absolute yield could not match between the simulation and observation though the interannual trend did.*

####################################################################

We described the detail of the input data and method (Section 4.2.1).

The error in the previous manuscript was caused by the following three points: (1) bias in shortwave radiation; (2) the large default value for the ratio of PAR to shortwave radiation; (3) our mistake in input data handling. After rectifying the three points, we recalculated rice yields over Japan. The revised result shows the error between simulations and observation is not large (Figure 13). However, the model still tends to overestimate the observations. One of the reasons for the simulation errors is thought to be the method of parameterisation. For the simulations over Japan, we used parameters parameterised for a variety "Koshihikari" only at one site in Japan (Section 3), although there is a large diversity in agricultural management and technique, and rice varieties planted throughout Japan. Therefore, parameterisation at a large scale is necessary for better large scale simulations. We added the discussion on the error to the revised manuscript (P7, L24-27).

Minor comments:

################################################################
*Section 5.2: Could you put the longname of 'RF' in MATCRO-RF?.*
################################################################
RF denotes "rain-fed". We described that MATCRO-RF denotes simulations under rainfed condisions (P8, L17-18).

################################################################
*Figure 5: What's the red curve? Clarify in the caption.*
################################################################
The red curve in Fig. 5 indicates the fitted curve used in the model. We added this description in the caption. The description for the red curve in Fig. 3 was also missing. Therefore, we added it.

################################################################
*Figure 9: Please make the range of y-axis smaller to show the fluctuation clearer.*
################################################################
We modified the range of y-axis of Fig. 9. The same modification was made in Fig. 16.

################################################################
*Figure 11,12,13: You may use the unit of [t ha-1] for easier recognition of values.*
################################################################
We modified the unit of Figures 11, 12 and 13 to [t ha-1].

[revised manuscript text omitted]